# HR+/HER2– Advanced Breast Cancer Treatment in the First-Line Setting: Expert Review

**Katarzyna J. Jerzak** [1]**, Nathaniel Bouganim** [2]**, Christine Brezden-Masley** [3]**, Scott Edwards** [4]**, Karen Gelmon** [5]**, Jan-Willem Henning** [6]**, John F. Hilton** [7] **and Sandeep Sehdev** [7,*]

1   Odette Cancer Centre, Sunnybrook Health Sciences, Toronto, ON M4N 3M5, Canada; katarzyna.jerzak@sunnybrook.ca
2   Cedars Cancer Centre, McGill University Health Centre, Montreal, QC H4A 3J1, Canada; nathaniel.bouganim@mcgill.ca
3   Marvelle Koffler Breast Centre, Mount Sinai Hospital, Toronto, ON M5G 1X5, Canada; christine.brezden@sinaihealthsystem.ca
4   Dr. H. Bliss Murphy Cancer Center, St. John's, NL A1B 3V6, Canada; scott.edwards@easternhealth.ca
5   Faculty of Medicine, University of British Columbia, Vancouver, BC V5Z 1M9, Canada; kgelmon@bccancer.bc.ca
6   Tom Baker Cancer Centre, Calgary AB T2N 4N2, Canada; janwillem.henning@albertahealthservices.ca
7   The Ottawa Hospital Cancer Centre, Ottawa, ON K1H 8L6, Canada; jfhilton@toh.ca
*   Correspondence: ssehdev@toh.ca

**Correction Statement:** This article has been republished with a minor change. The change does not affect the scientific content of the article and further details are available within the backmatter of the website version of this article.

**Abstract:** The approval of CDK4/6 inhibitors has dramatically improved care for the treatment of HR+/HER2– advanced breast cancer, but navigating the rapidly-expanding treatment evidence base is challenging. In this narrative review, we provide best-practice recommendations for the first-line treatment of HR+/HER2– advanced breast cancer in Canada based on relevant literature, clinical guidelines, and our own clinical experience. Due to statistically significant improvements in overall survival and progression-free survival, ribociclib + aromatase inhibitor is our preferred first-line treatment for de novo advanced disease or relapse >12 months after completion of adjuvant endocrine therapy and ribociclib or abemaciclib + fulvestrant is our preferred first-line treatment for patients experiencing early relapse. Abemaciclib or palbociclib may be used when alternatives to ribociclib are needed, and endocrine therapy can be used alone in the case of contraindication to CDK4/6 inhibitors or limited life expectancy. Considerations for special populations—including frail and fit elderly patients, as well as those with visceral disease, brain metastases, and oligometastatic disease—are also explored. For monitoring, we recommend an approach across CDK4/6 inhibitors. For mutational testing, we recommend routinely performing ER/PR/HER2 testing to confirm the subtype of advanced disease at the time of progression and to consider *ESR1* and *PIK3CA* testing for select patients. Where possible, engage a multidisciplinary care team to apply evidence in a patient-centric manner.

**Keywords:** advanced breast cancer; CDK4/6 inhibitors; HR+/HER2–; mutational testing; treatment considerations

## 1. Introduction

Breast cancer is the leading cause of cancer among Canadian women. One in eight Canadian women will develop breast cancer during their lifetime, and one in 34 will die from it. The Canadian Cancer Society estimated that 28,600 Canadian women would be diagnosed with breast cancer in 2022, representing 25% of all new cancer cases in women. It was also estimated that 5500 Canadian women would die from breast cancer in 2022, representing 14% of all cancer deaths in women for that year. On average, 78 Canadian women will be diagnosed with breast cancer every day [1].

Hormone receptor–positive (HR+), human epidermal growth factor receptor-2–negative (HER2–) breast cancer remains the most common subtype, accounting for ~80% of all breast

cancer cases [2]. Prior to 2016, the mainstay of treatment for HR+/HER2– advanced breast cancer in postmenopausal women was endocrine therapy (ET) either with tamoxifen, aromatase inhibitors (AIs; anastrozole, exemestane, or letrozole) or fulvestrant (combined with ovarian suppression in premenopausal women) [2]. The FALCON study demonstrated a progression-free survival (PFS) advantage—but not an overall survival (OS) advantage—for fulvestrant 500 mg versus anastrozole 1 mg in the first-line setting among patients with bone-only metastatic disease [3]. Combination treatment with fulvestrant and anastrozole achieved improved OS compared with anastrozole alone in a SWOG Cancer Research Network trial [4]. Furthermore, in the second-line metastatic setting, BOLERO 2 demonstrated a PFS advantage (but not an OS advantage) of adding the mTOR inhibitor everolimus to exemestane over exemestane alone [5].

Breast cancer is a heterogeneous disease, even within the various subtypes. In the last 10–15 years, the focus has been on developing biologically-directed therapies [2]. These include cyclin-dependent kinase 4 and 6 inhibitors (CDK4/6i), which emerged onto the Canadian breast cancer treatment landscape in 2016 with the approval of palbociclib for the first-line treatment of HR+/HER2– metastatic breast cancer in postmenopausal women [6], followed by ribociclib in 2018 [7] and abemaciclib in 2019 [8]. In 2020, ribociclib was also approved for first-line treatment of HR+/HER2– metastatic breast cancer in premenopausal women [9].

Since then, there has been a rapidly-expanding evidence base in HR+/HER2– breast cancer, including PFS and OS results and analyses of efficacy in specific patient populations, like those with early relapse or visceral metastases. Given the amount of evidence, distilling the data and the impact on practice pathways can be daunting.

Thus, the objective of this review is to provide oncologists with an expert opinion review of the existing evidence in HR+/HER2– metastatic breast cancer in the first-line treatment setting. We will also share some insights into our own practices and provide best practice recommendations where applicable.

## 2. Methods

We recognized that there is a need for a Canadian expert review paper in this area, as many medical oncologists who manage multiple tumor types often have questions about how to apply the clinical trial evidence to practice.

Thus, we gathered to discuss the topic and to review and agree on key questions being asked by medical oncologists. Thereafter, we searched and reviewed the relevant literature. The manuscript is structured using the clinical questions as subheadings. All authors contributed to and reviewed this manuscript.

As a guiding principle, we agreed that evidence from randomized controlled trials (RCT)—level 1 evidence—should guide treatment decisions and was prioritized in this expert review. Real-world evidence (RWE) can provide insights into sequencing decisions and safety and survival outcomes in populations who may not be represented in RCTs but are not designed to replace an RCT.

## 3. Results

### 3.1. Preferred Therapy for HR+/HER2– Advanced Breast Cancer

3.1.1. What Is the Preferred Treatment Combination in the First-Line HR+/HER2– Metastatic Setting?

Ribociclib + AI is the preferred treatment for patients with HR+/HER2– advanced breast cancer who have been either diagnosed de novo in the advanced setting or who have relapsed with the advanced disease more than 12 months after completing adjuvant ET for earlier stage disease (either tamoxifen or AI). While the general standard of care is the use of any CDK4/6i in combination with AI based on similar positive PFS in RCTs, only ribociclib has demonstrated OS benefit in the first-line setting to date.

The CDK4/6i were studied in combination with an AI for first-line use in HR+/HER2– patients with postmenopausal advanced breast cancer in PALOMA-2 (palbociclib + letro-

zole) [10,11], MONALEESA-2 (ribociclib + letrozole) [12], and MONARCH-3 (abemaciclib + nonsteroidal AI) [13]. Significant benefits in PFS—the primary endpoints—were demonstrated across all three trials. However, a significant OS benefit has only been demonstrated in MONALEESA-2, with a median OS of 63.9 months with ribociclib + letrozole versus 51.4 months with letrozole alone (hazard ratio (HR), 0.76; 95% confidence interval (CI), 0.63 to 0.93; *p* = 0.008) [14]. No OS benefit was demonstrated with palbociclib + letrozole compared to letrozole alone in PALOMA-2 (53.9 months vs. 51.2 months; HR, 0.956; 95% CI, 0.777 to 1.177; *p* = 0.3378) [15]. MONARCH-3 OS data are not yet mature, although an interim analysis suggested a trend of longer median OS with abemaciclib + nonsteroidal AI versus nonsteroidal AI alone (67.1 months vs. 54.5 months) [16]. Outcomes from these trials are summarized with the full PFS and OS curves (where available) in Figure 1.

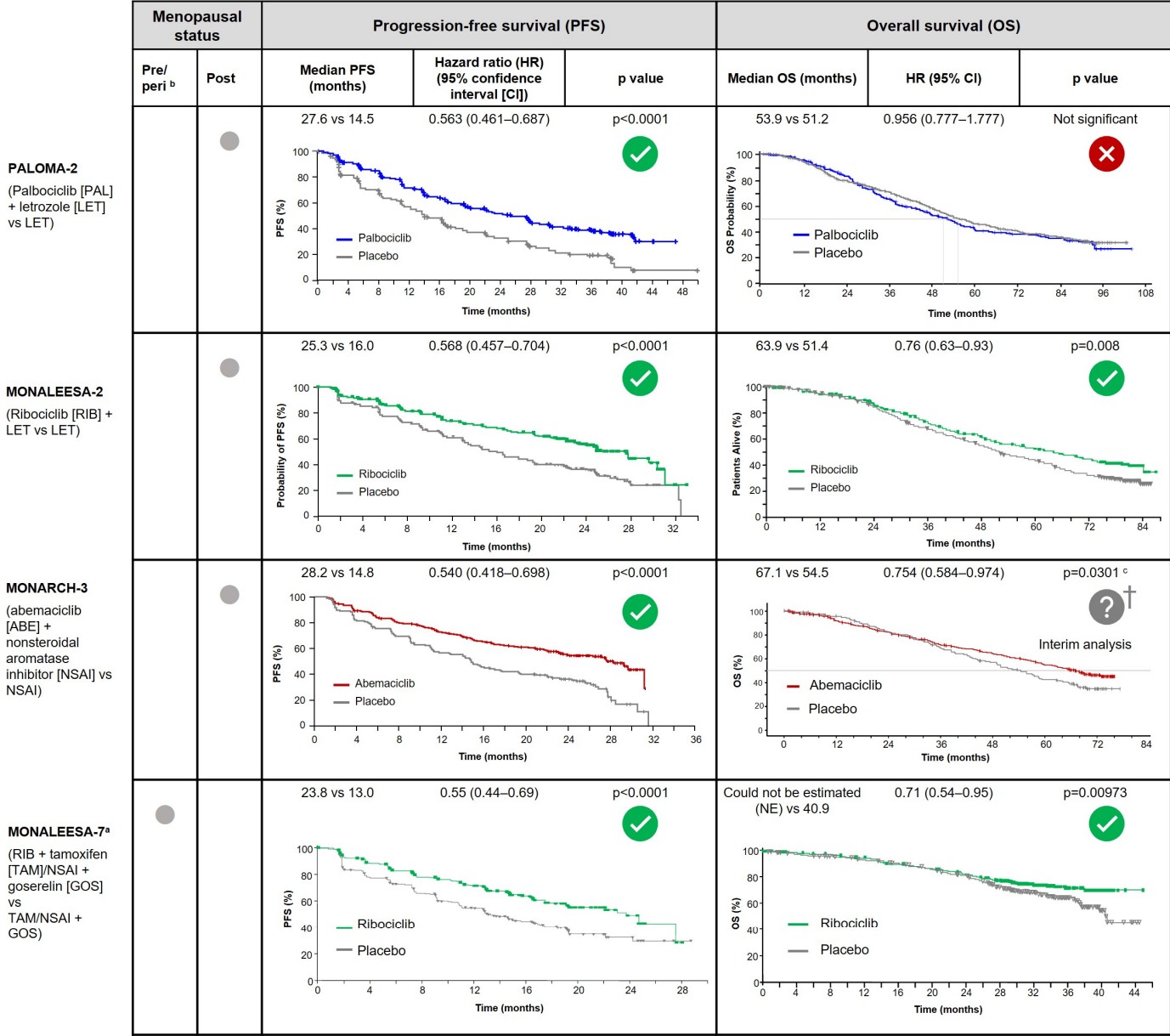

Patients had no prior treatment in the advanced breast cancer (aBC) setting, unless otherwise specified.
[a] Patients were allowed to have one line of chemotherapy in the advanced setting.
[b] With ovarian suppression
[c] Interim results

**Figure 1.** Summary of PFS and OS outcomes from phase 3 first-line CDK4/6i + nonsteroidal AI trials (investigator analysis).

In the premenopausal HR+/HER2– advanced breast cancer population, ribociclib + ET (in addition to ovarian ablation/suppression) is also preferred. Ribociclib + ET was studied in an exclusively pre/perimenopausal first-line trial (MONALEESA-7), demonstrating significantly longer PFS and OS compared to AI alone (outcomes also summarized in Figure 1) [17,18]. In contrast, as shown in Figure 2, abemaciclib and palbociclib pre/perimenopausal data come from subgroup analyses of the second-line MONARCH-2 and PALOMA-3 trials, with 17% and 21% of patients being pre/perimenopausal, respectively [19–22].

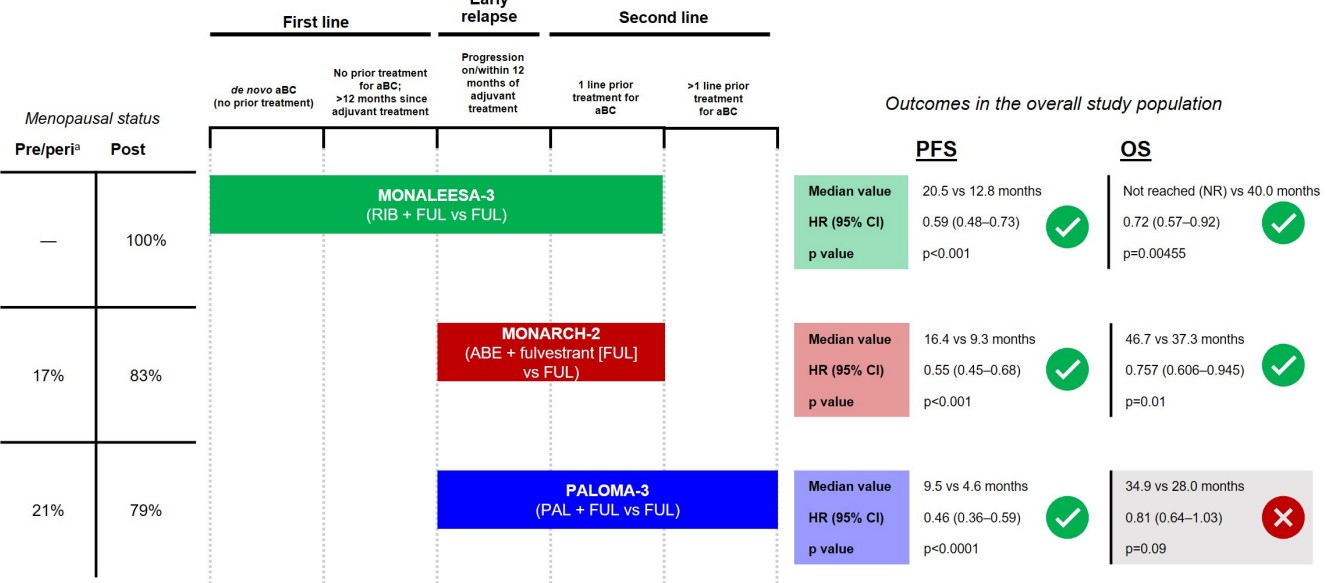

**Figure 2.** Summary of PFS and OS outcomes from phase 3 trials evaluating CDK4/6i + fulvestrant in HR+/HER2– advanced breast cancer.

Initially, it was thought that efficacy results for CDK4/6i were generalizable (i.e., demonstrated a "class effect") based on similar primary (PFS) outcomes. However, now that OS results are available, it is prudent to use agents that show a consistent survival advantage. A logical question emerges: why do these differences in OS exist? Notably, the median OS in the control arms of MONALEESA-2 and PALOMA-2 were 51.4 months and 51.2 months, respectively, suggesting these trial populations were similar, yet the median OS in the treatment arms were strikingly different (63.9 months and 53.9 months, respectively) [14,15]. Although trial differences (e.g., data loss in PALOMA-2 OS analysis, different rules for stopping due to toxicity) may have played some role, differences may also relate to the characteristics of each CDK4/6i. In terms of pharmacokinetics, the amount of free (unbound) drug available to penetrate tumors when given at the recommended starting dose is >22× higher with ribociclib compared to palbociclib in in vitro studies [23]. Abemaciclib and ribociclib are more active against CDK4 than CDK6, while palbociclib inhibits CDK4 and 6 equally [24,25]. This is relevant given that breast cancer cell lines have been associated with CDK4 gene dependency, while hematologic cell lines (i.e., that may be associated with neutropenia or other cell count abnormalities) were associated with CDK6 gene dependency [26]. Abemaciclib is the most potent CDK4/6i having the lowest IC$_{50}$ (the concentration at which 50% of the enzyme is inhibited), and it also has more off-target inhibition, including CDK1 and CDK9 [27,28]. These features may play a role in the different adverse event profiles (i.e., more GI toxicity) seen with abemaciclib and may also explain why abemaciclib is the only CDK4/6 with single-agent activity and continuous dosing [29].

Our preferred treatment is in accordance with current guideline recommendations from the National Comprehensive Cancer Network® (NCCN®), European Society for Medical Oncology (ESMO), and the 5th European School of Oncology (ESO)-ESMO international consensus guidelines for advanced breast cancer (ABC 5): CDK4/6i + AI or CDK4/6i + fulvestrant are preferred first-line therapies. Recommendations are summarized in Table 1. Given the absence of any head-to-head studies, all CDK4/6i are listed as preferred regimens. However, ribociclib + AI has a category 1 recommendation in the NCCN guidelines due to OS results, while abemaciclib + AI and palbociclib + AI are category 2A. In combination with fulvestrant, both ribociclib and abemaciclib have category 1 recommendations for first-line use, while palbociclib + fulvestrant is category 2A. Similarly, ribociclib + letrozole, ribociclib + fulvestrant, and abemaciclib + fulvestrant have ESMO clinical benefit scale (ESMO-MCBS) scores of 4 for first-line use in postmenopausal patients, while palbociclib + letrozole has a score of 3. Palbociclib + fulvestrant has a score of 4 for second-line use. In premenopausal patients, ribociclib + ET has an ESMO-MCBS score of 5 for first-line use, while no other CDK4/6i has a score assigned, given the absence of a dedicated trial [30–33].

**Table 1.** Current Guideline-Preferred First-Line ER+/HER2– Advanced Breast Cancer Regimens.

| ABC 5 * [30] | ESMO 2021 * [31] | NCCN (Version 3.2023) [33] |
|---|---|---|
| CDK4/6i combined with ET (AI or FUL) ESMO-MCBS v1.1 † scores: <br><br> • RIB + LET 1L (post): 4 <br> • PAL + LET 1L (post): 3 <br> • ABE + AI 1L (post): 3 <br> • RIB + FUL 1L, 2L (post): 4 <br> • ABE + FUL 1L, 2L (post): 4 <br> • RIB + ET 1L (pre): 5 | CDK4/6i combined with ET (AI or FUL): <br><br> • AI preferred if did not relapse on AI or relapsed >12 months of stopping adjuvant AI; otherwise, FUL preferred <br> • Comorbidities or performance status that precludes use of CDK4/6i combinations: ET alone <br> • See ABC 5 column for ESMO-MCBS scores | AI + CDK4/6i <br><br> • AI + RIB (category 1) <br> • AI + ABE (category 2A) <br> • AI + PAL (category 2A) <br> • FUL + CDK4/6i <br> • FUL + RIB (category 1) <br> • FUL + ABE (category 1) <br> • FUL + PAL (category 2A) |

* ABC and ESMO note the lack of head-to-head comparisons between CDK4/6i; direct cross-trial comparisons are not possible due to heterogeneous inclusion criteria. † Based on online scores at https://www.esmo.org/guidelines/esmo-mcbs/esmo-mcbs-scorecards accessed 1 February 2023. Cancer therapies are scored from 1–5 for therapies in the non-curative setting, where a score of 4 or 5 is considered to provide a substantial amount of clinical benefit. 1L, first line; 2L, second line; ABC, advanced breast cancer; ABE, abemaciclib; AI, aromatase inhibitor; CDK4/6i, cyclin dependent kinase 4/6 inhibitor; ET, endocrine therapy; ESMO, European Society for Medical Oncology; FUL, fulvestrant; LET, letrozole; MCBS, magnitude of clinical benefit scale; NCCN, National Comprehensive Cancer Network; NSAI, nonsteroidal aromatase inhibitor; PAL, palbociclib; pre, premenopausal; post, postmenopausal; RIB, ribociclib.

3.1.2. What If the Patient Relapsed on or within 12 Months after Completion of Adjuvant AI?

For patients progressing on or shortly after completion (≤12 months) of adjuvant AI ("early" relapse), our preference is to use ribociclib + fulvestrant or abemaciclib + fulvestrant.

Such patients may have more aggressive disease and/or some degree of endocrine resistance and a higher likelihood of having *ESR1* mutations, where estrogen receptor dysregulation may be more effective [34,35]. Thus, we prefer to switch to fulvestrant in combination with ribociclib or abemaciclib. The benefits of CDK4/6 inhibition in this setting have been dramatic.

CDK4/6i were studied in combination with fulvestrant in the MONALEESA-3 trial (ribociclib) [36], MONARCH-2 (abemaciclib) [19], and PALOMA-3 (palbociclib) [21]. Each trial included patients who progressed while on or within 12 months of completion of (neo)adjuvant ET (early relapse) and patients who progressed while receiving first-line ET for advanced breast cancer; additional patient populations included are shown in Figure 2. While no trial explicitly analyzed the subgroup of early relapse, the overall population and available subgroup analyses can inform treatment decisions in the early relapse population. Notably, MONALEESA-3 is the only one of the CDK4/6i + fulvestrant trials which included a population of patients with advanced breast cancer in the first line of treatment. Trial populations and outcomes are summarized in Figure 2.

In the overall population of MONALEESA-3, PFS and OS were significantly improved with ribociclib + fulvestrant versus fulvestrant alone (PFS: 20.5 months vs. 12.8 months; HR, 0.59; 95% CI, 0.48 to 0.73; $p < 0.001$; OS: not reached vs. 40.0 months; HR, 0.72; 95% CI, 0.57 to 0.92; $p = 0.00455$). These results were consistent in the early relapse + second line subgroup analysis (PFS: 0.565, 95% CI, 0.438 to 0.744; OS: 40.2 months vs. 32.5 months; HR, 0.73; 95% CI, 0.53 to 1.00). Data on early relapse patients alone are not available [36,37].

In the overall population of MONARCH-2, PFS and OS were significantly improved with abemaciclib + fulvestrant vs. fulvestrant alone (PFS: 16.4 months vs. 9.3 months; HR, 0.55; 95% CI, 0.45 to 0.68; $p < 0.001$; OS: 46.7 months vs. 37.3 months; HR, 0.757; 95% CI, 0.606 to 0.945; $p = 0.01$). Improvement was consistent in patients with primary ET resistance (i.e., relapsed during the first 2 years of [neo]adjuvant ET or progressed within the first 6 months of first-line ET for advanced breast cancer; PFS: HR, 0.454; 95% CI, 0.306 to 0.674; OS: 0.686; 95% CI, 0.451 to 1.043). However, these results were not stratified by whether the ET resistance was in the (neo)adjuvant (i.e., early relapse) or metastatic setting [19,20].

PALOMA-3 enrolled patients with advanced breast cancer who had disease progression after previous ET in the (neo)adjuvant or metastatic setting, including those who had received prior chemotherapy. Although the endpoint for PFS was met (9.5 months vs. 4.6 months; HR, 0.46; 95% CI, 0.36 to 0.59; $p < 0.0001$) [21], there was no statistically significant difference in OS with palbociclib + fulvestrant vs. fulvestrant alone (34.9 months vs. 28.0 months; HR, 0.81; 95% CI, 0.64 to 1.03; $p = 0.09$) [22]. Subgroup analyses are, therefore, not definitive but rather only hypothesis-generating.

In our practice, if the patient is experiencing early relapse but has only one or two metastatic sites restricted to the bone, ribociclib or abemaciclib may be combined with a different AI rather than fulvestrant. This is based on potential pragmatic drivers, such as preference for oral administration over intramuscular injection and potential funding implications of switching to fulvestrant in combination with CDK4/6i.

### 3.1.3. In Which Situations Should an Alternative to Preferred Therapy Be Used?

In some situations, alternatives to ribociclib + AI may be appropriate. Patients should be prescribed a CDK4/6i other than ribociclib if electrocardiogram (ECG) access is difficult, if patient compliance with ECG monitoring is problematic, or if concomitant medications that may also prolong QTc intervals cannot be changed. In such cases, abemaciclib would be preferred as it demonstrated a strong trend of prolonged OS at an interim MONARCH-3 analysis. Where access to abemaciclib is not possible or its potential toxicity unacceptable, palbociclib would be a reasonable choice given its PFS benefit in the absence of demonstrated OS.

In some cases, patients may not be suited for treatment with a CDK4/6i + AI. Medications that significantly affect CYP3A4 metabolism may be problematic, and medication review by oncology pharmacists should be considered prior to initiation. Furthermore, patients who cannot or choose not to adhere to monitoring requirements for CDK4/6i use (e.g., due to geography or temperament) may be better suited for et al. one [38,39], particularly with fulvestrant if they have bone-only disease [3,40]. However, because the long-term benefits of CDK4/6i are dramatic, we recommend offering the preferred therapy with the appropriate safeguards.

In addition to the above situations, the contraindications, as noted in the respective product monographs, are to be observed. Abemaciclib, palbociclib, and ribociclib all cite hypersensitivity to the drug or any ingredient in the formulation as a contraindication [41–43]. Ribociclib also cites untreated congenital long QT syndrome, a QTcF interval of $\geq$450 msec at baseline, and those who are at significant risk of developing QTc prolongation as contraindications [43].

### 3.1.4. What Phenotyping or Mutational Testing Should Be Undertaken at the Time of Progression?

Where feasible, we recommend obtaining a biopsy at the time of diagnosis of metastatic disease to reassess the phenotype (ER, PR, and HER2). We recommend that HER2 status be quantified by immunohistochemistry (IHC) and not simply reported qualitatively as amplified or not amplified.

In the case of patients who relapse early, *ESR1* mutational analysis of the metastatic site may be warranted, particularly if an AI backbone is being considered. The SoFEA and EFECT trials investigated the second-line use of fulvestrant versus exemestane in patients with advanced HR+ breast cancer who had previously progressed on a nonsteroidal AI. A combined analysis demonstrated that, at baseline, 30% of patients were positive for *ESR1* mutations, which were associated with shorter PFS. However, among patients with an *ESR1* mutation, significantly longer PFS was observed with fulvestrant compared to exemestane (3.9 months vs. 2.4 months; HR, 0.59; 95% CI, 0.39 to 0.89; *p* = 0.01) [35].

Monitoring *ESR1* mutations is important because early targeting with a switch from AI backbone to fulvestrant can result in clinical benefit. In the open-label PADA-1 phase 3 trial, patients with HR+/HER2– breast cancer receiving palbociclib7 + AI were monitored for rising levels of *ESR1* mutations using circulating tumor DNA (ctDNA) and were randomized to either continue with their current treatment or switch to palbociclib (same dose) + fulvestrant. Median PFS was significantly longer in patients who switched to palbociclib7 + fulvestrant compared to those who stayed on their original therapy (11.9 months vs. 2.7 months; HR, 0.61; 95% CI, 0.43 to 0.86; *p* = 0.0040) [44]. However, this does not necessarily suggest that ctDNA assessment should currently become routine clinical practice. Of note, in the United States, elacestrant has been approved for use in *ESR1* mutation–positive HR+/HER2– advanced or metastatic breast cancer with disease progression following at least one line of ET, with NCCN guidelines recommending one line of prior therapy containing a CDK4/6i [33].

*PIK3CA* mutational analysis may also be warranted, either from a progressive metastatic site or archival tissue (since it does not usually change with disease progression), to guide future consideration of the option of fulvestrant with alpelisib [31] or clinical trials of newer targeted agents, such as AKT inhibitors [45].

High microsatellite instability (MSI) is rare in ER+ metastatic breast cancer, and the role of MSI and tumor mutational burden (TMB) in guiding the use of immunotherapy remains unclear. Other alterations (such as *ERBB2*, *AKT1*, *MYC*, *CCND1*) may indicate a worse prognosis, though guidelines do not recommend routine testing since their presence would not currently change treatment [46]. We do not routinely test for these alterations in our practice. Wherever possible, interested and eligible patients should be considered for applicable clinical trials.

Though a discussion of second-line treatment is beyond the scope of this article, it is noteworthy that there is a rapidly evolving field of research quantifying HER2 expression, including HER2-low and HER2-ultralow. This may affect the planning of post-CDK4/6i treatment options for patients classified as HER2-low, given the efficacy of trastuzumab deruxtecan in this population.

A proposed flow chart for mutational testing is outlined in Figure 3.

### 3.1.5. What Are Best Practices for Monitoring Patients on CDK4/6i?

We recommend a standard monitoring approach based on product monographs to help manage adverse reactions associated with CDK4/6i, using temporary treatment interruptions and/or dose reductions as needed. The most common adverse reactions associated with CDK4/6i (≥20% with any CDK4/6i in first-line trials in combination with a nonsteroidal AI and greater than placebo) are shown in Figure 4, and notably include blood cell count abnormalities (i.e., neutropenia, leukopenia, anemia), abnormal liver functions tests, and diarrhea. QTc prolongation has also been reported with ribociclib. As such, monitoring complete blood count, ECG, liver function, and diarrhea is important. Each CDK4/6i

product monograph includes specific guidance on monitoring (Table 2). For example, palbociclib only includes dose adjustment recommendations for hematologic toxicities (i.e., neutropenia) and nonhematologic toxicities in general; in contrast, abemaciclib and ribociclib include specific dose modification recommendations for hematologic toxicities, liver toxicity, and interstitial lung disease (ILD)/pneumonitis, among others (described in Table S1) [41–43]. We suggest a simplified approach to monitoring, described below and summarized in Figure 5.

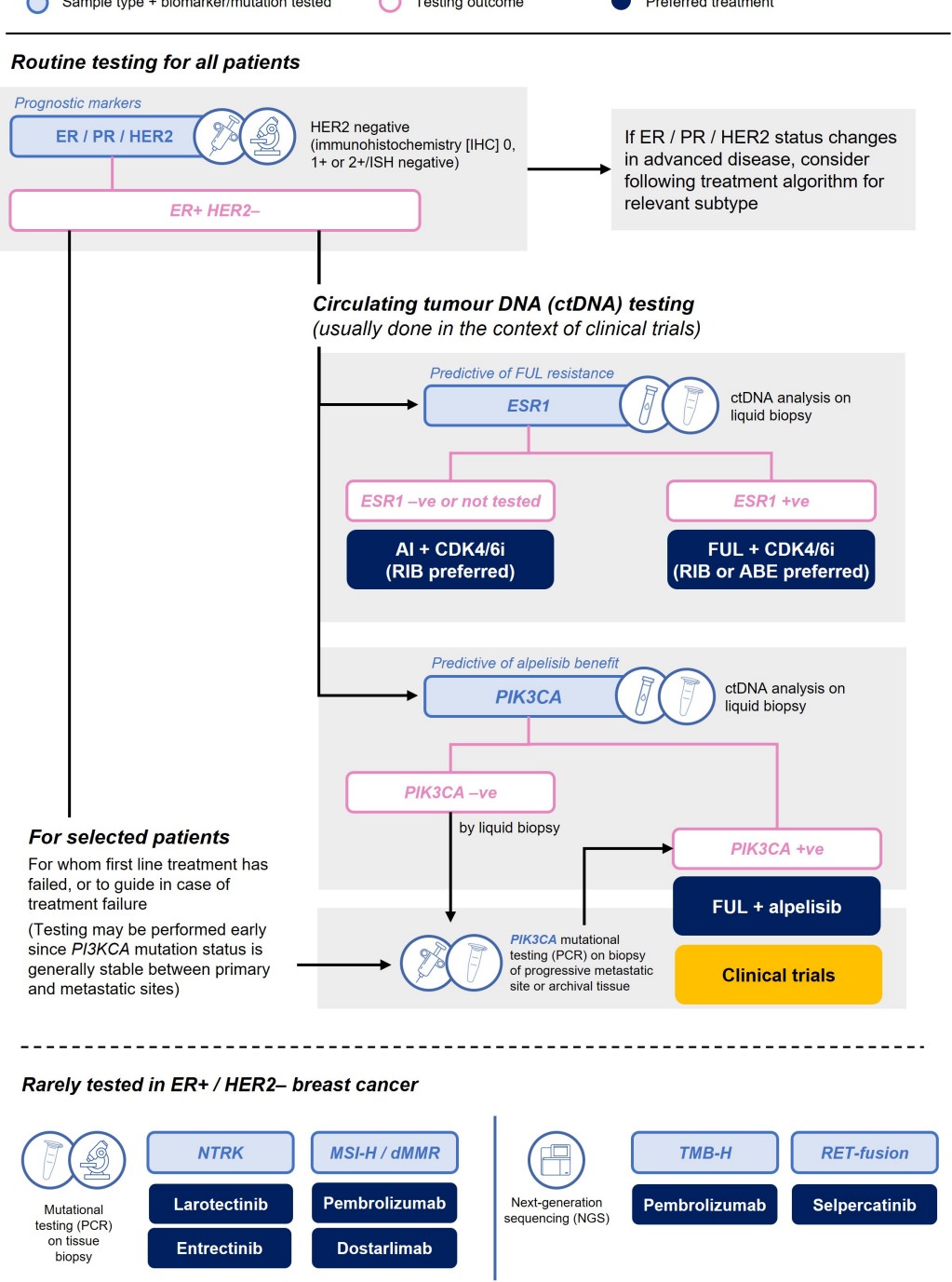

**Figure 3.** Mutational testing at diagnosis of advanced/metastatic disease.

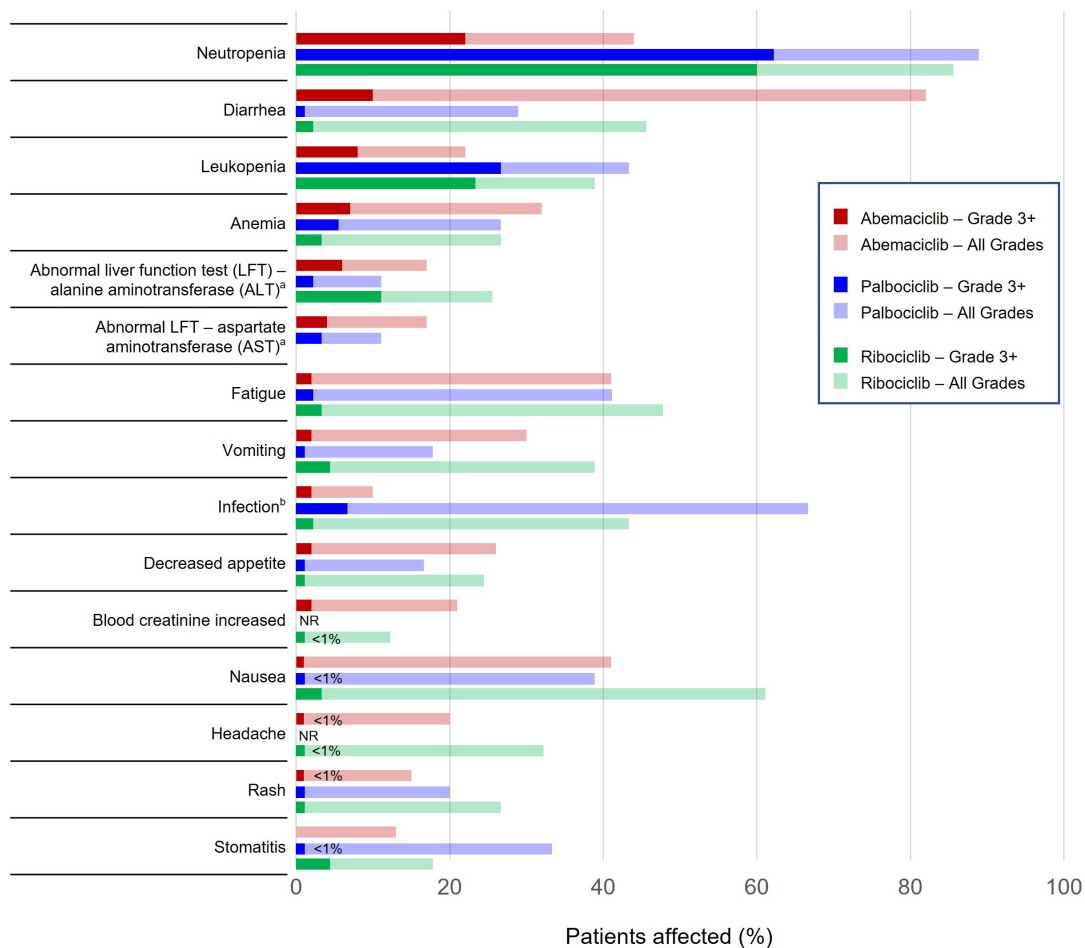

**Figure 4.** Incidence of most common adverse reactions as per product monograph in patients with advanced breast cancer receiving CDK4/6i + nonsteroidal AI.

NR, not reported.
[a] Ribociclib data are for combined ALT and AST abnormal LFTs.
[b] Rates of highest-incidence single types of infections summed for all Grade (Grade 3+) for abemaciclib (lung: 7% [2%]; upper respiratory tract infection: 10% [0]) and ribociclib (upper respiratory infection: 39% [2%]; urinary tract infection: 19% [1%]).

**Table 2.** Monitoring as per Recommendations in Product Monographs.

|  | **Abemaciclib [41]** | **Palbociclib [42]** | **Ribociclib [43]** |
|---|---|---|---|
| CBC | Prior to start<br>Q2W for months 1, 2<br>QM for months 3, 4<br>As clinically indicated | Prior to start<br>Day 15 for cycle 1, 2<br>As clinically indicated | Prior to start<br>Q2W for cycles 1, 2<br>At the start of cycles 3–6<br>As clinically indicated |
| Liver function | Prior to start<br>Q2W for months 1,2<br>QM for months 3,4<br>As clinically indicated | Monitor for signs of hepatoxicity | Prior to start<br>Q2W for cycles 1, 2<br>At the start of cycles 3–6<br>As clinically indicated |
| ILD/pneumonitis | Monitor for pulmonary symptoms indicative of ILD/pneumonitis (hypoxia, cough, dyspnea) | | |
| ECG | – | – | Prior to start<br>Day 14 of cycle 1<br>Beginning of cycle 2<br>At regular intervals during steady state (approx. cycle day 14) & as clinically indicated |

**Table 2.** *Cont.*

| | Abemaciclib [41] | Palbociclib [42] | Ribociclib [43] |
|---|---|---|---|
| Electrolytes | – | – | Prior to start<br>At regular intervals & as clinically indicated |
| Infection or myelosuppression | | Monitor for signs and symptoms | |
| Thrombo-embolism | Monitor for signs and symptoms | – | Monitor for signs and symptoms in patients at risk |

CBC, complete blood count; ECG, electrocardiogram; ILD, interstitial lung disease; LFT, liver function tests; PE, pulmonary embolism; Q2W, every two weeks; QM, every month; QTc, corrected QT interval; VTE, venous thromboembolism.

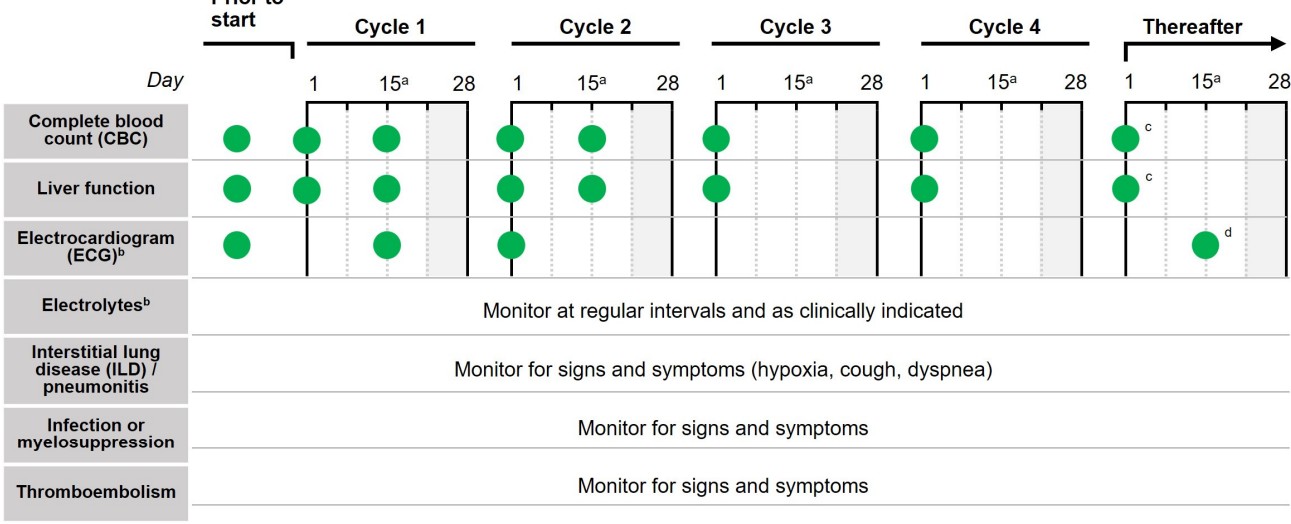

a Slight variations in the product monographs (PMs) re: mid-cycle testing point (e.g., monitoring on Day 14, 15, or Q2W; continuing for 2 or 4 cycles after the initial 2)
b Based on ribociclib PM
c May continue on Day 1 of Cycles 5–6, or as clinically indicated
d At regular intervals (~cycle Day 14), and as clinically indicated

**Figure 5.** Suggested monitoring approach for CDK4/6i.

- Complete Blood Count

Complete blood counts should be performed prior to initiating CDK4/6i therapy, every two weeks for the first two cycles, at the beginning of the next two to four cycles, and as clinically indicated thereafter (e.g., after re-initiation of therapy upon resolution of neutropenia). If absolute neutrophil count (ANC) is below $1000/mm^3$ (Grade 3 neutropenia [47]) interrupt therapy until levels are recovered to $1000/mm^3$ and start the next cycle at the same dose. If levels drop below $1000/mm^3$ again or below $500/mm^3$ at any time, withhold therapy until levels are recovered to $1000/mm^3$ and resume at the next lower dose. Many experienced clinicians may opt to continue CDK4/6i therapy even if the ANC is slightly lower than $1000/mm^3$. Adjusting cycle intervals is also an option for some patients.

Febrile neutropenia, defined as Grade 3 neutropenia with a single episode of fever >38.3 °C or a sustained temperature above 38 °C for more than one hour [47] has been reported in 1–3% of patients on CDK4/6i [41–43]. In such cases, the medication should be stopped, and the patient evaluated for signs and symptoms of infection. Patients should be instructed to promptly report any fever to their cancer center.

- Electrocardiogram

Cardio-oncology is a relatively new discipline. According to the European Society of Cardiology, "the overarching goal of the cardio-oncology discipline is to allow patients with cancer to receive the best possible cancer treatments safely, minimizing cancer treatment-related cardiovascular toxicities (CTR-CVT)" [48]. Many Canadian cancer centers have integrated the services of a cardio-oncology team into their cancer clinics.

Per the product monographs, ribociclib requires ECG monitoring. Discontinuations due to QTcF prolongation occurred in 0.6% of patients receiving ribociclib + fulvestrant as noted in MONALEESA-3 study [36]. No specific rate of discontinuation was reported in other trials, but in MONALEESA-2, 11 patients (3.3%) in the ribociclib group had at least one average QTcF interval longer than 480 msec after baseline—including 1 patient with baseline cardiac abnormalities and 6 with an increase greater than 60 msec from baseline. Most were able to continue treatment (600 mg ribociclib) without interruption [12]. Across phase 3 trials, 1.4% of patients receiving ribociclib had dose interruption, reduction, or discontinuation [49].

Experts in the field often take a similar approach across all CDK4/6i. Based on guidance for ribociclib, complete a baseline ECG prior to initiating therapy (QTcF should be <450 ms), at Day 14 of cycle one, and at the beginning of the second cycle, and then at the discretion of the clinician on or around Day 14 of the cycle. Serum electrolytes should similarly be assessed prior to initiating therapy and at regular intervals throughout treatment, correcting abnormalities before initiation or continuation. Review of all concomitant medications should also be standardized to ensure no concern for risk of QTc prolongation [42].

In the case of a QTcF above 480 ms or recurrent QTcF greater than or equal to 481 ms (including events >500 ms), interrupt dosing until prolongation resolves to less than 481 ms, and then reinitiate therapy at the next lower dose. In the event of Torsade de Pointes, polymorphic ventricular tachycardia, unexplained syncope, or signs/symptoms of serious arrhythmia, permanently discontinue therapy.

More frequent monitoring is recommended in patients with underlying risk factors for Torsade de Pointes, receiving concomitant drugs that prolong the QTc interval, or if QTc prolongation is detected during ribociclib treatment [43].

- Liver Function Tests

Like the complete blood count, liver function testing—including alanine transaminase (ALT), aspartate transaminase (AST), and bilirubin levels—should be performed prior to initiating therapy, every two weeks for the first two cycles, at the beginning of the next two to four cycles, and as clinically indicated thereafter. If AST and/or ALT levels are recurrently/persistently >3–5× the upper limit of normal (ULN) or >5–20× ULN at any time, in the absence of bilirubin increases above 2× ULN, withhold therapy until levels recover to ≤3× ULN and resume at the next lower dose. If AST and/or ALT is above 20× the ULN or bilirubin is >2× ULN in the absence of cholestasis, discontinue treatment. Monitoring recommendations are based on the abemaciclib and ribociclib product monographs.

- Diarrhea

Start antidiarrheal treatment (e.g., loperamide and increase oral fluid intake) at first sign of diarrhea. If an increase of 4–6 stools/day over baseline does not resolve to fewer than 4 stools/day over baseline within 24 h, suspend treatment until resolution then resume without dose reduction. If toxicity persists after resuming therapy and with maximal supportive measures, if there is an increase of ≥7 stools/day over baseline, or in cases of hospitalization or life-threatening consequences, hold dose until toxicity resolves and resume at the next lower dose. Monitoring recommendations are based on the abemaciclib product monograph [41,47]. In real-world settings, patients may not accept chronic grade 1 diarrhea and dose reduction may also be appropriate [41]. Thankfully, tachyphylaxis often develops over the first 2–3 weeks.

- Other

Interstitial lung disease (ILD)/pneumonitis has occurred and patients on CDK4/6i should be monitored for indicative pulmonary symptoms, such as hypoxia, cough, and dyspnea. For symptomatic ILD/pneumonitis (Grade 2), interrupt therapy until asymptomatic and resume at the next lower dose. In the event of severe ILD/pneumonitis, discontinue the CDK4/6i. Monitoring recommendations are based on the ribociclib product monograph. Recommendations are consistent in the abemaciclib product monograph with the caveat that dose interruption/modification is only required for Grade 2 toxicity if persistent or recurrent that does not resolve with maximal supportive measures within 7 days.

Patients at risk of thromboembolic events should be closely monitored during therapy. In the case of Grade 3 or 4 thromboembolic events, suspend dosing and treat appropriately, resuming CDK4/6i once the patient is clinically stable. Monitoring recommendations are based on the abemaciclib product monograph.

It may also be prudent to monitor for mouth soreness/stomatitis, which may be a sign of infection. For other toxicities in general, Grades 1 and 2 do not require dose adjustment. For Grade 3 or 4 toxicities with abemaciclib or palbociclib, withhold therapy until resolved then resume at the next lower dose. For Grade 3 toxicities with ribociclib, suspend dose until toxicity resolves and resume at same dose (initial case) or next lower dose (recurrent); for Grade 4 toxicities, discontinue ribociclib.

Monitoring circulating tumor DNA (ctDNA) has been shown to detect disease progression well in advance of radiological assessment in both patients responding and not responding to CDK4/6i [50]. Thus far, there is no evidence that early change in therapy would result in improved long-term outcomes. Currently, there is no public funding for such assessments.

### 3.2. Special Populations

### 3.2.1. What about Preferred Treatments for 'Fit' Elderly Patients?

In our practices, age is not a deciding factor in selecting first-line treatment in HR+/HER2– advanced breast cancer. Assessing fitness (or frailty) is more important than age. An elderly person with Eastern Cooperative Oncology Group Performance Status (ECOG PS) of 0–1 who is "fit" should be prescribed the standard of care treatment of CDK4/6i + AI. In most cases, ribociclib in combination with AI remains the preferred CDK4/6 agent based on the totality of the evidence (PFS, OS, and quality of life [QoL]). The caveat to this approach might be the patient with limited life expectancy where single-agent ET might be preferred.

Some tools used to assess frailty include Cancer and Aging Research Group (CARG) score (validated for prediction of chemotherapy toxicity) [51], Vulnerable Elders Survey (VES-13) [52], comprehensive geriatric assessment (CGA) [53], Geriatric-8 [54], Groningen frailty score [55], and Clinical Frailty Scale [56] (see Table 3), but none have been validated as predictive of toxicity or outcomes for CDK4/6i.

**Table 3.** Format and Clinical Utility of Tools Used to Assess Frailty.

| Tool | Format | Clinical Utility |
|---|---|---|
| CARG [57] | Physician-administered; assesses 11 clinical parameters | Used to assess toxicity risk in geriatric patients receiving chemotherapy |
| VES-13 [52] | Patient-reported; 13-item function-based assessment | Detects vulnerable older adults who are at an increased risk of death or functional decline over the subsequent 2 years |
| CGA [53] | Physician-administered; assesses functional status, comorbidities, polypharmacy, cognition, psychological status, social support and nutrition | Used to identify changes that are potentially treatable to improve patient outcomes |
| Geriatric 8 [54] | Physician-administered; 8-item screening tool | Specific to cancer patients; rapid (5 min) |

**Table 3.** *Cont.*

| Tool | Format | Clinical Utility |
|---|---|---|
| Groningen [55] | Patient-reported; 15-item self-assessment | Reflects current problems in a patient's daily life |
| Clinical Frailty Scale [56] | Physician-administered; based on observation and clinical judgement (not a questionnaire) | Not specific to oncology; requires clinical judgement |

On occasion, the decision may be made to start unfit patients on a lower initial dose of a CDK4/6i. Such a decision may be based on experience or a "gut feeling" (i.e., the art of medicine), rather than on specific evidence. This is a reasonable approach for such patients but should not be adopted as standard practice for elderly patients. Furthermore, increasing to the standard dose should be attempted.

The AMALEE trial demonstrated that a reduced starting dose of ribociclib (400 mg per day orally for 3 weeks with 1 week off) was not noninferior to the usual dose (600 mg per day orally for 3 weeks with 1 week off) though it was associated with less change in QTcF from baseline [58].

A pooled analysis of PALOMA phase 2 and 3 studies evaluated the hematologic adverse events following palbociclib dose reduction from 125 mg to 100 mg (per day orally for 3 weeks on, 1 week off). The lower dose reduced the frequency and severity of hematologic adverse events. Most dose reductions occurred within the first 3 months of palbociclib treatment. This analysis did not evaluate efficacy [59].

A retrospective, single-center, chart review evaluated the effect of reduced dose intensity in the first 12 weeks of palbociclib treatment. Patients with low dose intensity (defined as taking <80% of the prescribed doses) experienced a significantly shorter PFS at 12 weeks than patients with high dose intensity (defined as taking >80% of the prescribed doses). At 36 weeks there was a similar trend, though not statistically significant [60].

For many targeted therapies in oncology, dose reductions for toxicity have not been found to compromise efficacy. However, that does not imply that expected benefits would be observed if starting all patients a priori on lower doses. Toxicity may be a marker of pharmacokinetics. In our practice, we strive to initiate all patients on the recommended dose; but on occasion we may start a patient on lower doses for reasons of tolerability and then try to work up to standard doses thereafter.

3.2.2. What about Preferred Treatments for Patients with Visceral Disease?

Historically, there was a perception that chemotherapy was necessary in patients with HR+/HER2– advanced cancer with visceral disease. However, the best treatment for endocrine-sensitive disease is ET with or without CDK4/6i, not chemotherapy [30]. The populations of all the first-line landmark trials included ~50% patients with visceral disease and all demonstrated a significant improvement in PFS in this population (Table 4). Furthermore, a network meta-analysis compared the HRs for PFS of CDK4/6i + ET, ET alone, and chemotherapy. No chemotherapy regimen—with or without targeted therapy—was significantly better than CDK4/6 inhibitors + ET in terms of PFS [61].

Rarely, a patient may have visceral crisis, defined as severe organ dysfunction and rapid progression of disease with impending organ failure [30]. Such cases are rare and warrant the most rapidly efficacious therapy [30]. Given recent evidence with CDK4/6i in aggressive advanced disease, it is unclear that chemotherapy would be better than ET + CDK4/6i; though if chemotherapy is used, doublets should be considered.

The RIGHT Choice trial compared ribociclib + AI to investigator's choice doublet chemotherapy (docetaxel + capecitabine; paclitaxel + gemcitabine; or capecitabine + vinorelbine) in pre/perimenopausal women with aggressive HR+/HER2– advanced breast cancer [62]. Approximately 60% of enrolled patients were first diagnosed in the advanced setting (i.e., de novo disease), ~60% of patients had symptomatic visceral metastases, and about half of patients had visceral crisis at baseline. First-line ribociclib + AI achieved

a statistically significant PFS benefit of ~1 year over combination chemotherapy in this patient population with aggressive disease (median PFS of 24 months vs. 12.3 months; HR, 0.54; 95% CI, 0.36 to 0.79). Ribociclib + AI had a similar objective response rate (ORR) and time to response (TTR) compared to doublet chemotherapy. This trial may not be reflective of Canadian practice due to the choice of doublets in the comparison arm.

**Table 4.** PFS in Subgroup of Patients with Visceral Disease in First-Line HR+/HER2– Trials.

| Study | Subgroup | PFS HR (95% CI) |
|---|---|---|
| PALOMA-2 (49% visceral) | Site of metastases at baseline<br>Visceral<br>Nonvisceral | <br>0.63 (0.47–0.85)<br>0.50 (0.36–0.70) |
| MONALEESA-2 (59% visceral) | Liver or lung metastases<br>Yes<br>No | <br>0.561 (0.424–0.743)<br>0.597 (0.426–0.837) |
| MONARCH-3 (53% visceral) | Metastatic site<br>Visceral<br>Bone only<br>Other | <br>0.61 (0.42–0.87)<br>0.58 (0.27–1.25)<br>0.34 (0.19–0.61) |
| MONALEESA-7 (57% visceral) | Liver or lung metastases<br>Yes<br>No | <br>0.50 (0.38–0.68)<br>0.64 (0.45–0.91) |

CI, confidence interval; HR, hazard ratio; PFS, progression free survival.

Earlier in this review, we discussed how CDK4/6i + fulvestrant is the preferred treatment for early relapse—patients who experience relapse with advanced disease on or less than 1 year after completion of adjuvant ET. Some clinicians may opt to treat early relapse with chemotherapy for 1–2 cycles to achieve a rapid response before starting fulvestrant and CDK4/6i therapy. There is no specific evidence to support this practice and none of the first-line postmenopausal landmark CDK4/6i trials allowed for such an algorithm, though the RIGHT Choice trial may provide some insight into this practice. MONALEESA-7 in pre/perimenopausal women did, however, allow one cycle of chemotherapy in the advanced setting.

3.2.3. What about Practical Considerations for Other Special Populations, like Those with Brain Metastases or Those with Oligometastatic Disease?

Breast cancer with central nervous system (CNS) metastases has a poor prognosis. Pre-clinical studies indicate that CDK4/6i may have activity in the CNS [63,64]. The Canadian abemaciclib product monograph states that "concentrations of abemaciclib and its active metabolites in cerebrospinal fluid are comparable to unbound plasma concentration" [42], suggesting that it does cross the blood brain barrier [65]. However, the first-line trials discussed previously either did not enroll patients with brain tumors or enrolled too few to draw any conclusions on clinical efficacy of CDK4/6i in patients with brain metastases. A nonrandomized phase II trial (NCT02308020) of abemaciclib investigated intracranial objective response rate (iORR) in subtype-specific cohorts. In a cohort of patients with brain metastases secondary to HR+/HER2– breast cancer (*n* = 58), the primary endpoint was not met (iORR of 5.2%; 95% CI, 0.0 to 10.9) [66]. The intracranial clinical benefit rate was 24% (95% CI, 13.1 to 35.2) and median OS was 12.5 months (95% CI, 9.3 to 16.4). A volumetric decrease in target intracranial lesions was experienced by 38% of patients. Prospective trials are ongoing to investigate the efficacy in brain metastases of abemaciclib (NCT02308020), palbociclib (NCT02896335 and NCT04334330), and abemaciclib plus stereotactic radiosurgery (SRS) vs. palbociclib plus SRS vs. ribociclib plus SRS (NCT04585724).

There is a subset of patients with metastatic breast cancer who present with limited disease, termed 'oligometastatic' breast cancer [67]. Oligometastatic breast cancer is defined

as metastatic disease with limited spread and thus it may be possible to achieve longer survival in this select patient population [67]. Local techniques of surgery and radiotherapy (RT) are often employed with oligometastatic disease [67], though further discussion of these treatments is beyond the scope of this paper. Combining systemic therapies, such as CDK4/6i, with RT for oligometastatic disease is an area of interest for many researchers. CDK4/6i can contribute to repair of DNA double-strand breaks and as such may enhance the effect of RT [67]. A small number of preliminary studies suggest that the combination of CDK4/6i and RT is safe, especially for bone metastases, with comparable hematologic toxicity [67]. Additional studies with larger populations and longer follow-up will be needed before the significance for clinical practice is known.

### 3.3. Patient Considerations

What Is Important to Patients When Considering Treatment Options?

We know from our patients with metastatic breast cancer that they want to live longer, and they want that longer life to be of good quality [68]. Thus, we strongly recommend that clinicians inform patients of the overall survival evidence supporting the various agents so that they can make an informed decision about their treatment.

According to the Canadian Breast Cancer Network (CBCN) survey results (see Figure 6), OS, PFS, and QoL are the most important factors for patients [68]. QoL assessments have been performed as part of some of the pivotal clinical trials. However, incorporating QoL assessments into clinical decision-making is difficult due to variability in data collection and rating scales used among the trials and the subjectivity of these assessments. Our perspective on QoL data is that evidence should demonstrate that the medication does not have a negative impact. In other words: if a medication is safe, has shown benefit with respect to overall survival, and shows no detriment to QoL, it is a reasonable choice.

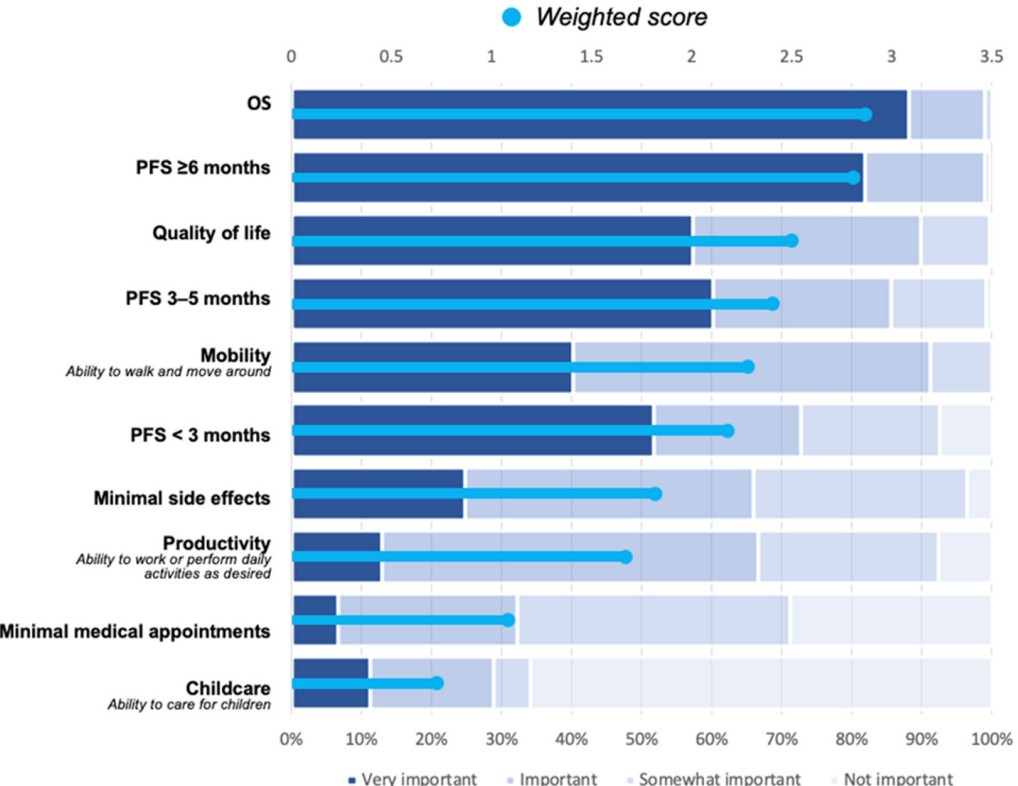

**Figure 6.** Importance of different treatment characteristics in decisions for metastatic breast cancer patients (Canadian Breast Cancer Network).

*3.4. The Breast Cancer Team*

Which Allied Health Care Partners Can Play a Role in the Care of Patients with Advanced Breast Cancer?

HR+/HER2– advanced breast cancer can be treated by medical oncologists without a multidisciplinary team. That said, allied health care partners are of great assistance and can be leveraged where possible.

- Pharmacy

Pharmacists are important members of the oncology team who help ensure safe and appropriate drug use. They can have active roles in side effect monitoring and treatment, dose modifications, patient education, and reimbursement navigation. Adherence is critical for optimal breast cancer treatment outcomes, and pharmacists can be instrumental in facilitating adherence in collaboration with the patient and caregiver. Ideally, pharmacists with knowledge and experience in oncology should be involved.

Pharmacists are well-placed to provide recommendations on drug interactions. They evaluate the potential for interactions between prescribed medications and other supportive care medications and foods. They can provide patient guidance for mitigation of clinically significant drug interactions and collaborate with prescribers to discuss interaction management strategies when necessary [69].

Consider discussing the clinical verification of prescriptions with community and hospital pharmacists and as needed, suggesting the use of appropriate tools. For example, Cancer Care Ontario has developed a checklist to define quality in cancer prescription dispensing in pharmacies. This includes validating basic patient details, checking for previous adverse effects, identifying barriers to adherence, ensuring that the dose and regimen are correct—considering any necessary dose modifications or drug-drug interactions—and, that the patient is receiving appropriate supportive care, education, and monitoring. The complete tool is shown in Figure 7.

- Nursing

Oncology nurses, "pivot" nurses, or nurse practitioners can be of great assistance to coordinate laboratory and ECG monitoring, to inquire about side effects, and to assist patients in understanding the dosing schedules of their oral medication. They offer unique skill sets with respect to symptom management, emotional support, and coordination of other ancillary and supportive therapies.

- General practice

In some cases, the medical oncologist may wish to enlist the help of the general practitioner who often has a long-term relationship with the patient and may be in a unique position to provide reassurance, counseling support, advance directive planning, and assistance in symptom management and palliation.

- Mental health

Patients with breast cancer may face mental health challenges as they adjust to their illness or treatment, including grief, depression, and anxiety. Patients should be routinely screened for distress using a simple tool or questionnaire. Patients experiencing distress should be offered psychological intervention by qualified individuals, such as coping skills and communication training, support groups, individual counseling, and family/caregiver counseling. Patients with signs of anxiety or depression should be referred to a psychiatrist, psychologist or other psychosocial specialist [33,70].

Complementary therapies, such as mediation, mindfulness, relaxation, and creative therapies (e.g., music, art) are recommended for patients experiencing distress; and exercise has also been shown to have positive effects on physical health. Practical problems, such as financial or career concerns can be managed with the assistance from social work services.

An illustration of how the multidisciplinary team functions in managing the care of the patient with advanced breast cancer is seen in Figure 8. This will differ by center depending on the available resources.

## Clinical Verification of Cancer Drug Prescriptions

**Patient**

- ❑ Verify patient using two identifiers present on Rx
- ❑ Confirm height and weight on Rx with patient
- ❑ Check for allergies
- ❑ Confirm diagnosis/indication with patient
- ❑ Identify if new or continuing treatment
- ❑ Check for toxicity or intolerance from previous cycle (if applicable)
- ❑ Identify barriers to adherence

**Regimen**

**For the regimen, verify correct:**
- ❑ Indication
- ❑ Drugs & route
- ❑ Scheduling & interval
- ❑ Start date correct interval from previous treatment (if applicable)

**Dose**

- ❑ Verify correct dose for indication
- ❑ Verify correct calculated dose for patient using BSA and/or weight (if applicable)
- ❑ Check for modified dose (if applicable)
- ❑ Check for drug interactions

**Patient Care**

- ❑ Verify supportive care provided
- ❑ Identify what education has been provided and reinforce
  - how and when to administer
  - cycle schedule
  - importance of adherence
  - proper handling, storage and disposal
  - side effects and management strategies
- ❑ Identify toxicities to monitor & plan follow-up

See Cancer Care Ontario's Drug Formulary website for drug and regimen monographs: www.cancercare.on.ca/toolbox/druaformulary

**Figure 7.** Clinical verifications of prescriptions checklist tool for community pharmacists.

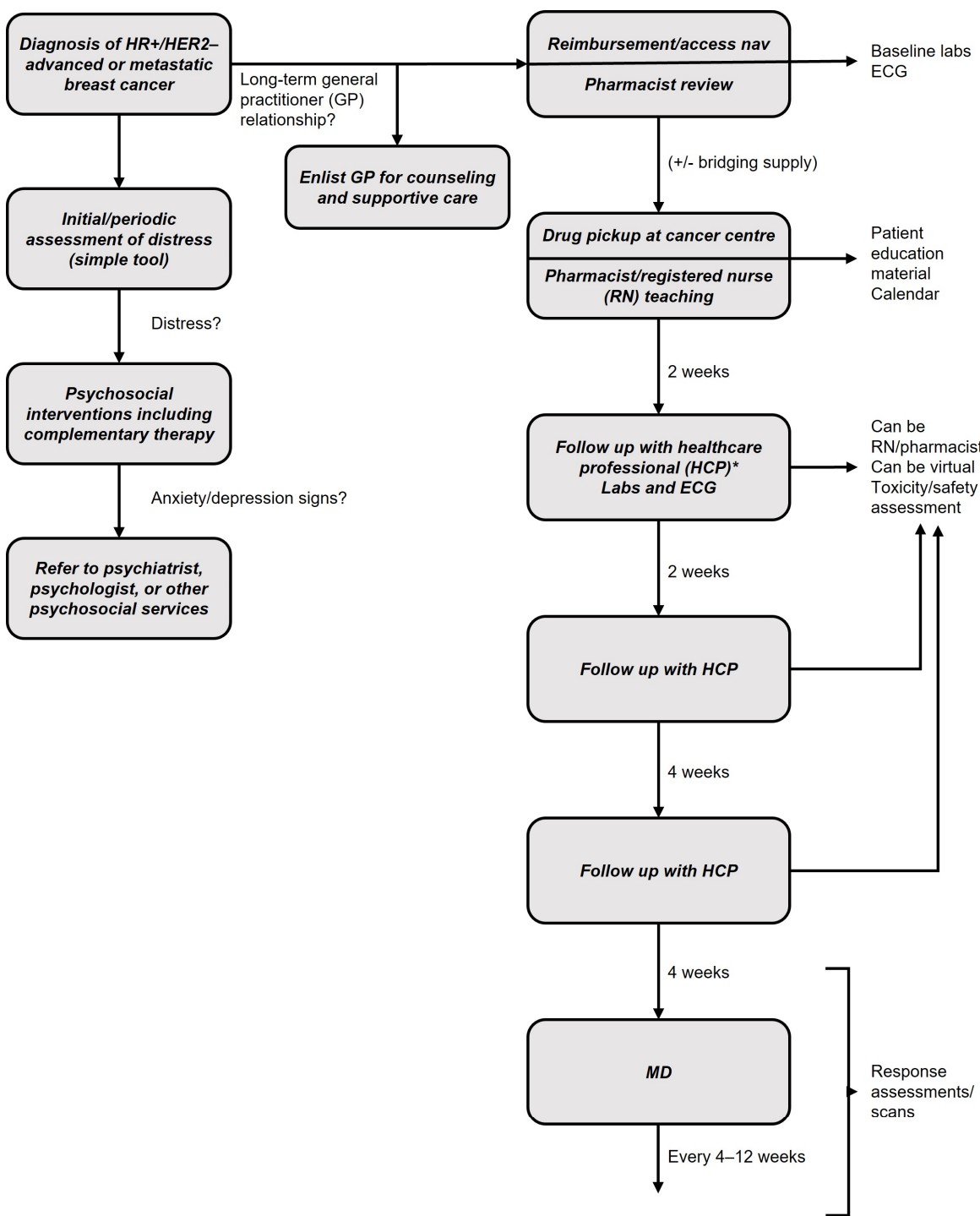

*Many clinics may have toxicity/tolerance assessments done by oral therapy oncology pharmacists or RNs.

**Figure 8.** Summary of preferred treatment pathway for first-line HR+/HER2– advanced breast cancer.

## 4. Practical Application and Future Directions

Based on the evidence of current randomized phase III trials and our expert opinion in applying that evidence to practice, we have summarized the standard of care and preferred treatments for HR+/HER2– advanced breast cancer for patients relapsing more than and less than 12 months after completion of adjuvant ET, for patients with visceral metastases, and for frail patients. A summary of our recommendations can be found in Figure 9.

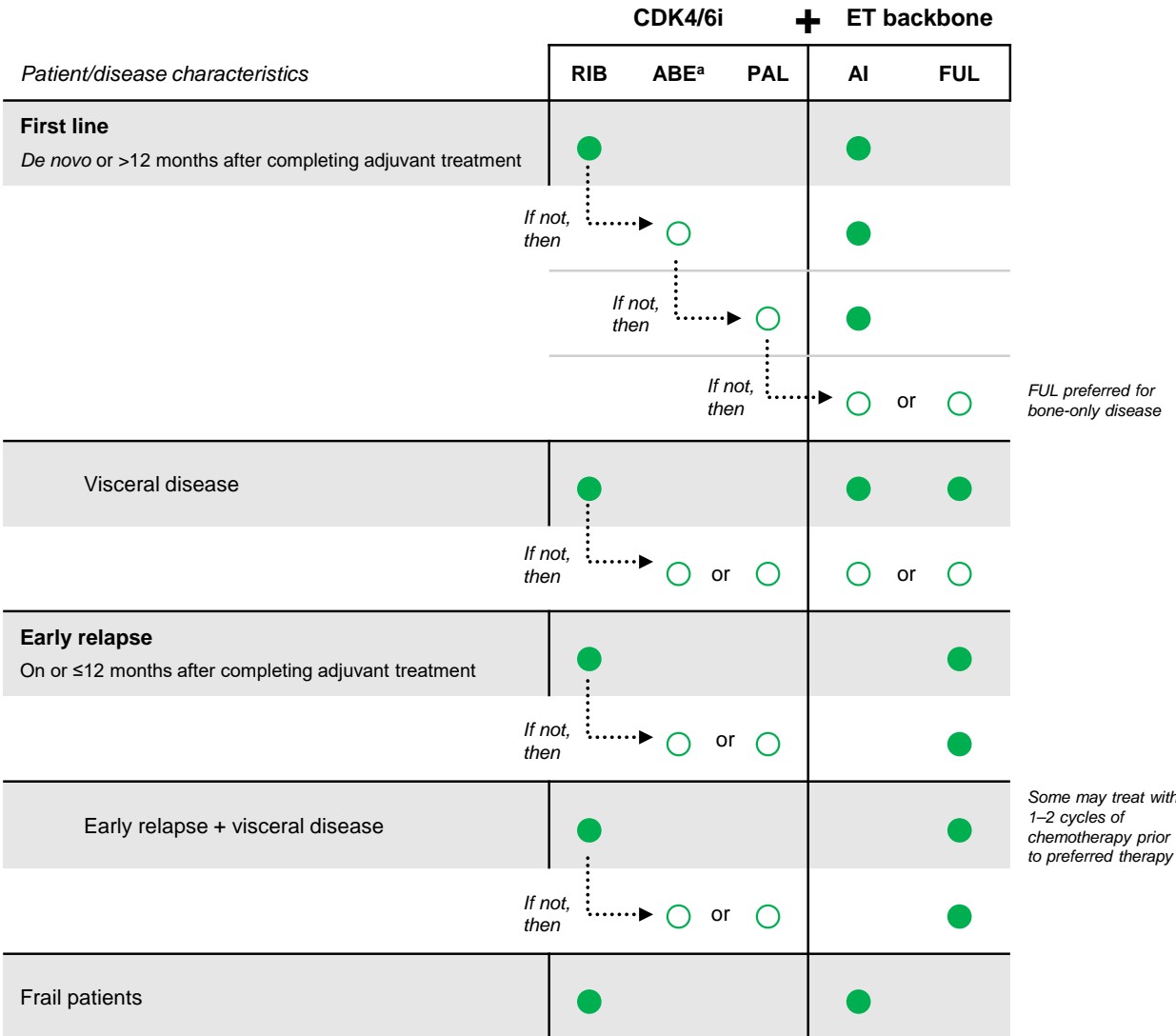

<sup>a</sup> At the time of publication, there is no public funding for abemaciclib in the metastatic setting in Canada.

**Figure 9.** Summary of preferred first-line treatments for HR+/HER2− advanced breast cancer.

With regards to future directions, further understanding of biomarkers and gene signatures is essential to better understand tumour biology and individualize therapy for patients. Additionally, oral selective estrogen receptor degraders (SERDs) are soon to become available in Canada and will find their place in the treatment of HR+/HER2− advanced breast cancer. Our focus in this article is first line treatment of HR+/HER2− in the advanced setting with current available data. However, there are many clinical trials of interest investigating how to treat progression on first line treatments (MAINTAIN NCT02632045, PACE NCT03147287 and postMONARCH NCT05169567). Additionally, the use of antibody conjugates, such as trastuzumab deruxtecan, is of great interest in breast cancer in the HER2 positive, as well as HER2-low and HER2-ultralow metastatic breast cancer populations.

**5. Conclusions**

Research in breast cancer has led to remarkable progress in the understanding and management of metastatic breast cancer. With new treatments and a wealth of clinical trial data, clinicians treating patients with advanced HR+/HER2− breast cancer must discuss such evidence with their patients and apply evidence to practice.

**Supplementary Materials:** The following supporting information can be downloaded at: https://www.mdpi.com/article/10.3390/curroncol30060411/s1, Table S1. Dose Modification for Specific Adverse Reactions Listed in at Least One CDK4/6i Product Monograph.

**Author Contributions:** Conceptualization, K.J.J., N.B., C.B.-M., S.E., K.G., J.-W.H., J.F.H. and S.S.; Writing—original draft, K.J.J. and S.S.; Writing—review and editing, K.J.J., N.B., C.B.-M., S.E., K.G., J.-W.H., J.F.H. and S.S.; Development of figures, K.J.J., S.S. and C.B.-M.; Supervision, K.J.J. and S.S. All authors have read and agreed to the published version of the manuscript.

**Funding:** Financial support for research of this article was provided in the form of an unrestricted educational grant from Novartis Pharmaceuticals Canada Inc. to liV Agency Inc.

**Acknowledgments:** We would like to thank Novartis Pharmaceuticals Canada Inc. for supporting the development of this manuscript through an unrestricted educational grant to liV Agency Inc. We thank liV Agency Inc. for logistical and organizational support and for medical writing support (Iris Boraschi, Janice Carr Meisner, Katherine Conti, David Haberl, Ian Hellstrom, Lisa Kellenberger).

**Conflicts of Interest:** K.J.J.: Advisory boards—Amgen, AstraZeneca, Apo Biologix, Eli Lilly, Esai, Genomic Health, Gilead Sciences, Knight Therapeutics, Merck, Myriad Genetics Inc, Pfizer, Roche, Seagen, Novartis, Viatris. Speaker honoraria—Amgen, AstraZeneca, Apo Biologix, Eli Lilly, Esai, Genomic Health, Gilead Sciences, Knight Therapeutics, Merck, Myriad Genetics Inc, Pfizer, Roche, Seagen, Novartis, Viatris. Grants/research support—AstraZeneca, Eli Lilly, Seagen; N.B.: Honoraria—Merck, Novartis, Astra Zeneca; C.B.-M.: Advisory boards/consultancy and speaker honoraria*—Astra Zeneca, Agendia, BMS, Knight, Merck, Lilly*, Novartis*, Pfizer*, Roche, Seagen, Taiho, Sanofi, Mylan, Gilead; S.E.: Apobiologix, Astellas, Astra Zeneca, Bayer, Eli Lilly, Gilead; K.G.: Advisory boards—Astra Zeneca, Merck, Gilead, Seagan, Pfizer, Novartis, Lilly; J.-W.H.: Advisory boards and speaker honoraria—Astra Zeneca, Novartis, Pfizer, Gilead, Eli Lilly, Seagen, Merck; J.F.H.: TBD; S.S.: Advisory board and speaker honoraria—Astra Zeneca, Novartis.

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
