# Peer review of "HR+/HER2– Advanced Breast Cancer Treatment in the First-Line Setting: Expert Review"

_curroncol, doi:10.3390/curroncol30060411_

Round 1
Reviewer 1 Report
In this narrative review, the authors provide the recommendations for the first-line treatment of HR+/HER2- advanced breast cancer, based on the clinical trial results, relevant literature, clinical guidelines and their clinical experience. It is a well-written and well-organized review on this topic and the Tables/Figures are well-illustrated and relevant. This review article would add a great value to the current literature and help the breast cancer care team provide best practice to the patient.
Author Response
See attached file. Thank you for your review.

Reviewer 2 Report
The manuscript by Jerzak and colleagues is a narrative comprehensive review on the use of CDK4/6 inhibitors for the treatment of patients with HR+ HER2- advanced BC. The manuscript is well written, comprehensive, and it may be a good reference in daily practice for both experienced and new fellow oncologists.
After some minor revisions, this author suggests this manuscript for publication.
- Firstly, most of the figures need to be revised. They appear blurry, and overall the quality is not - satisfactory.
- When discussing the blood exams, it would be good to assess febrile neutropenia, adding treatment approach and possible differential diagnosis (e.g. excluding infections, etc).
- Similarly, when discussing liver functions test, it would be useful to add few words to suggest differential diagnosis (e.g. liver progression)
- In paragraph 3.2.2, a network metanalysis published in 2019 (PMID: 31494037) may be probably helpful
Author Response
Thank you for your review. Please see attached file.

Reviewer 3 Report
Comments:
a) All figures except for figure 2 seem to be of poor resolution. Please consider uploading better quality images.
b) The authors should consider adding ctDNA analysis for patient monitoring when possible. There is evidence that rising ctDNA in patients on CDK4/6i could be used to identify progression early in advanced disease. (PMID: 35201661) Possibly at the end of cycle3 in patients showing poor response?
c) The authors could also mention monitoring of additional alterations in ERBB2, AKT1, MYC, CCND1 when possible in addition to ESR1, PI3KCA mutations that were mentioned.
Author Response
Thank you for your review. Please see attached file

Reviewer 4 Report
The authors provide an expert opinion review on how to treat HR+/HER2- advanced breast cancer patients in the first-line setting. This sound Expert Review is well structured and organized, it contributes significantly to the field and it is excellent described. Figures and Tables summarize and highlight perfectly the data reported and discussed. This review will help considerably to oncologists and other specialists (pharmacists) to establish right treatments against patients suffering from breast cancer.
Author Response

(The authors gave the same response as above.)
